# OpenReview forum: "PrivAct: Internalizing Contextual Privacy Preservation via Multi-Agent Preference Training"
_ICML.cc/2026/Conference — ICML 2026 regular_

### Official Review · Reviewer_AeeR · 2026-03-01

**Soundness:** 3
**Presentation:** 3
**Significance:** 3
**Originality:** 3
**Overall Recommendation:** 4
**Confidence:** 2

**Summary:**

The paper investigated the problem of contextual privacy in multi-agent preference learning.  The authors proposed a method called "PrivAct" that propagates privacy-aware rewards through a tree-structured multi-agent generation process and optimizes agents via preference learning with leakage-conditioned asymmetric reward shaping.  Based on the method, the authors did comprehensive experiments to evaluate multiple properties such as the privacy-helpfulness tradeoff, transferability, robustness, and so on.

**Compliance With Llm Reviewing Policy:**

Affirmed.

**Final Justification:**

The rebuttal from the authors solved my concerns. I raised my score from 3 to 4.

**Key Questions For Authors:**

1. What information are you protecting for privacy precisely in your problem? And provide a real-world motivation for which part is sensitive, correspondingly, and why it is necessary to protect it.

2. Are the metrics for privacy and helpfulness (such as privacy score and helpfulness score) used in previous work? If yes, please cite them.

3. What is the key technical part to help you achieve privacy-preserving, and what is the novelty compared with previous work?

**Limitations:**

yes

**Strengths And Weaknesses:**

Strengths:

1. The paper studied an interesting and important problem.

2. The authors did comprehensive empirical experiments.

Weaknesses:

1. The writing is unclear and confusing.  The paper claims contextual privacy preserving in the title. However, in the main text, there is confusion about which part is sensitive information since the authors mentioned "privacy-preserving outcomes",   "privacy preference," and so on. It is unclear which part the authors are protecting for privacy.

2. Lacking theoretical guarantees makes the paper not convincing and solid enough.  The paper lacks rigorous theoretical results and specific mathematical definitions, such as for "topologies".

---

> ### Author Rebuttal · Authors · 2026-03-30
>
> > W1. Confusion about which part is sensitive information
> >
> > Q1. What information are you protecting for privacy precisely in your problem?
>
> 1. Definition and scope of contextual privacy
>
> In section 3.1 the contextual privacy this paper is protecting is defined with respect to a set of sensitive information items $\mathcal{S}$. The goal is: given a context and a user instruction, generate a response that avoids disclosing information in $\mathcal{S}$. This aligns with recent works on contextual privacy, where LLMs may unintentionally leak privacy at inference time due to unawareness of the privacy norms governing appropriate information flow, even without malicious attackers [1].
>
> 2. Concrete examples of protected information
>
> As an example, an LM agent shares information that John is “talking to a few companies about switching jobs” when assisting John in sending an email to John’s manager. Similarly, Fig 4 in our paper illustrates the type of information we aim to protect: a congregant (Mark) inquires with a clergy (Jane) about community well-being, and the clergy (Jane) is expected to generate a response while holding confidential information regarding another congregant (Sarah).
>
> 3. Positioning within the broader privacy landscape
>
> As summarized in Appendix A.1, our work focuses on implicit/contextual privacy. This is distinct from (i) explicit privacy (e.g., PII), (ii) long-term memory privacy (e.g., RAG-based leakage), and (iii) model privacy (e.g., model inversion attacks). This taxonomy helps disambiguate our scope and situates our contribution within the broader privacy literature.
>
> We will revise the paper by adding a concise clarification in Sec. 1 to preempt potential confusion.
>
> > W2. Lacking rigorous theoretical results and specific mathematical definitions, such as for "topologies".
>
> First, in our multi-agent system, agents generate responses sequentially, each taking actions conditioned on the shared context. This naturally induces a per-step preference structure, where at each step the model distinguishes between better and worse continuations given the current prefix. Leveraging Theorem 6.1 from [2], DPO applied to such per-step preference pairs is equivalent to optimizing an advantage-weighted RL objective. This establishes a principled connection between our method and RL, providing theoretical support for our training framework.
>
> Second, regarding multi-agent topology, we will clarify that our system can be formalized as a directed acyclic graph (DAG) $G = (V, E)$, where each node $a_i \in V$ represents an agent, and each directed edge $(a_i, a_j) \in E$ indicates that the output of $a_i$ is provided as input to $a_j$. A complete generation trajectory corresponds to a path through $G$, ending at a terminal node $a_N$ that produces the final response $r_N$.
>
> > Q2. Are the metrics for privacy and helpfulness used in previous work? If yes, please cite them.
>
> The metrics are adopted from PrivacyLens [1]. Specifically,
> - The privacy score is defined as 1 - LR, where LR (Leakage rate) is defined and used in [1],
> - The helpfulness score used in our work is a normalized version of the raw score defined in [1].
>
> We’ve included appropriate citations in Section 4.2.
>
> > Q3. What is the key technical part to help you achieve privacy-preserving, and what is the novelty compared with previous work?
>
> We thank the reviewer for this question.
>
> Our approach achieves privacy-preserving behavior through 2 technical components. First, we curate a multi-agent contextual privacy preference dataset and introduce a distributed multi-agent training method. Second, to optimize the privacy-helpfulness tradeoff we propose leakage-conditioned asymmetric reward shaping (LC-ARS). This mechanism discourages the shortcut of trading small privacy violations for large utility gains, guiding the model to learn behaviors that are both helpful and privacy-compliant.
>
> Compared to prior work, our novelty lies in these aspects. First, internalizing contextual privacy. We move beyond inference-time prompt based privacy enhancement or external safeguards by training models to internalize contextual integrity and act accordingly, enabling generalizable behavior and avoiding leakage surface expansion. Table 1 compares existing approaches and illustrates our motivation. Second, we introduce a contextual privacy-aware multi-agent training framework and a reward shaping mechanism LC-ARS specifically designed for the privacy-helpfulness trade-off.
>
>
> [1] Shao, Yijia, et al. "Privacylens: Evaluating privacy norm awareness of language models in action." Advances in Neural Information Processing Systems 37 (2024).
>
> [2] Setlur, Amrith, et al. "Rl on incorrect synthetic data scales the efficiency of llm math reasoning by eight-fold." Advances in Neural Information Processing Systems 37 (2024).

---

> > ### Author Rebuttal · Reviewer_AeeR · 2026-04-02
> >
> > My concerns are solved, and I raised my score.

---

> > > ### Author Response · Authors · 2026-04-03
> > >
> > > We sincerely thank the reviewer for the positive feedback. We are glad that our response addressed your concerns.

---

### Official Review · Reviewer_nFzu · 2026-03-08

**Soundness:** 3
**Presentation:** 2
**Significance:** 3
**Originality:** 3
**Overall Recommendation:** 4
**Confidence:** 4

**Summary:**

This paper proposes **PrivAct**, a training framework for multi-agent LLM systems that internalizes contextual privacy preservation directly into the agents’ generation policies. Instead of relying on inference-time safeguards such as prompts or external controllers, the method uses preference optimization to train agents to produce responses that respect contextual privacy while remaining helpful. The framework combines multi-agent generation with reward propagation and a reward design that prioritizes privacy preservation when optimizing responses. Experiments across several open-source LLM backbones show improved privacy–helpfulness tradeoffs on the PrivacyLens benchmark and demonstrate zero-shot transfer to the ConfAIde probing benchmark.

**Compliance With Llm Reviewing Policy:**

Affirmed.

**Final Justification:**

The rebuttal substantively addressed my main concerns, and I am raising my score to a weak accept.

My primary concern (W2/Q1) was the lack of hyperparameter sensitivity analysis for α and β in LC-ARS. The authors provided ablations over six configurations spanning both concave and convex regimes, showing that nearly all settings strictly dominate Vanilla and PPE baselines — this substantially increases confidence in the design. My second concern (W3/Q3) regarding LLM judge reliability was addressed via GPT-4o cross-validation, yielding consistent conclusions, and further supported by the previously reported human-annotation agreement (Fleiss' κ = 0.82). The additional evaluation on the compositional privacy benchmark strengthens generalization claims beyond the primary benchmarks.

Residual limitations remain — inapplicability to closed-source models, English-only benchmarks, and an underdeveloped limitations section — but these are inherent to the training-based paradigm rather than fundamental flaws. The contribution is original, technically sound given the additional evidence, and significant for privacy-aware agentic systems.

**Key Questions For Authors:**

1. **Justification on the choice of α, β values**: The shaping exponents α=0.5 and β=2.0 are fixed across all experiments, but changing their values — particularly setting α ≥ 1 or β < 1 — would qualitatively alter the optimization regime. Yet, the fixed choice was not justified or evaluated in the paper. Could the authors provide results for at least two alternative (α, β) settings (e.g., α=0.8 and α=1.0 with β=2.0) to demonstrate that the two-regime structure, rather than the specific exponent values, drives the gains? Without this, it is unclear whether the results depend critically on this particular hyperparameter configuration. This is the most important question: a convincing sensitivity analysis would substantially strengthen confidence in the LC-ARS design.

2. **Hyperparameter sensitivity**: In Fig 2, the proposed method is sensitive to the choice of b for some models. b=0.45 has much worse helpfulness than b=0.3. Please explain the sensitivity and add more experiments on different b to reveal the whole picture of the hyperparameter sensitivity.

3. **LLM judge consistency**: The Mistral-7B leakage judge is applied to evaluate all methods, including during preference data generation. Could the authors provide a consistency analysis — for example, comparing Mistral-7B judge labels against a different evaluator (e.g., GPT-4o or human annotation) on a sample of 100–200 test responses — to validate that the judge's leakage assessments are reliable and not systematically affected by evaluation biases? This would directly validate an important assumption underlying all reported results.

4. **Reward propagation design choice**: The credit assignment strategy of averaging children's values for intermediate agents (Section 3.2.2) is a specific design choice. Could the authors briefly justify why expected-value averaging is preferred over alternatives such as minimum-value propagation (which would penalize any trajectory through a leaking child) or discounted propagation? A note on this choice would help readers understand the design space.

**Limitations:**

The paper includes an Impact Statement that briefly acknowledges responsible deployment considerations. However, the limitations discussion should be expanded to discuss the following:

- **Computational cost**: Fine-tuning multiple agents per backbone requires substantial compute. This is not discussed despite being practically relevant.
- **Closed-source model inapplicability**: The approach requires model weight access. This constraint is significant for practitioners using GPT-4, Claude, or Gemini.
- **Evaluation scope**: The benchmarks are English-only; cross-lingual generalization is neither assessed nor mentioned as a limitation.
- **LLM judge reliability**: The evaluation relies on LLM-as-a-judge for both leakage and helpfulness. Limitations of this approach — including potential false negatives on subtle contextual leakage — should be acknowledged.

The limitations discussion is insufficient and needs expansion.

**Strengths And Weaknesses:**

### Strengths

* Soundness: The reward design enforces helpfulness only when no leakage occurs, discouraging privacy–utility tradeoffs; consistent improvements across multiple model families and scales, along with ablations and transfer results, support robustness.
* Presentation: The paper is clearly organized, and the privacy–helpfulness Pareto visualization effectively communicates tradeoffs; figures and appendix materials provide sufficient implementation details.
* Significance: The work targets contextual privacy in multi-agent LLM systems, an underexplored but practically important problem; eliminating inference-time prompt engineering improves robustness and deployment practicality.
* Originality: The approach uses preference-based training to internalize contextual privacy for agent actions; treating privacy as a prerequisite rather than a competing objective is a novel design choice.


### Weaknesses

- **Clarity**: Some minor issues in the presentations.
  * Why is the privacy-helpfulness trade-off considered a challenge to tackle? I did not find prior literature mentioned in the paper to justify the significance of the trade-off challenge. For example, other trade-offs, like privacy-task-completion-rate, could be
  * The experiment section lacks essential details about the used dataset and tasks. The section only briefly mentioned the name of the dataset and scale, without mentioning the tasks in the dataset. Is this dataset a math problem-solving or something else? Lacking the context makes the meaning of "helpfulness" vague, and therefore, it makes less sense to evaluate helpfulness.

- **[Major] LC-ARS hyperparameter sensitivity unverified**: The shaping exponents α=0.5 and β=2.0 in Equation 1 are fixed across all experiments with no ablation. These are not merely scaling parameters: α controls whether the leakage penalty is concave (α < 1) or convex (α > 1), which qualitatively changes how small-leakage events are penalized. Setting α ≥ 1 could remove the disproportionate small-leakage discouragement that the paper identifies as a key design property. Similarly, β < 1 would change the non-leaking regime from convex to concave, altering the high-quality helpfulness incentive. The paper does not demonstrate that its results are robust to these choices. An ablation over α ∈ {0.3, 0.5, 0.8, 1.0+} and β ∈ {1.5, 2.0, 3.0} is needed to support the claim that LC-ARS works beyond this one hyperparameter setting. Without it, it's unclear whether the gains come from the specific exponents or from the two-regime structure.

- **[Moderate] LLM-as-a-judge reliability warrants validation**: The leakage judge J_L is Mistral-7B-Instruct-v0.2 (Appendix C.2), which is used both to generate preference training data and to evaluate all methods at test time. While the judge is applied equally to all methods (which prevents it from selectively favoring PrivAct), systematic biases in the judge — such as false negatives on subtle, indirect contextual leakage — would affect reported privacy metrics for all methods. Cross-model consistency across Llama and Qwen backbones provides indirect support that results are not artifacts of judge biases, but an explicit consistency analysis comparing Mistral-7B judge labels against an independent evaluator on a sample of test responses would directly validate this. The authors should provide such an analysis or acknowledge the limitation.

- **[Minor]** The approach requires model weight access for fine-tuning, limiting applicability to closed-source models (GPT-4, Claude, Gemini) that are commonly used in deployed multi-agent systems. While this is a limitation shared by all learning-based methods, it should be acknowledged explicitly in the paper's limitations section.

---

> ### Author Rebuttal · Authors · 2026-03-30
>
> > W1. Clarity
>
> The trade-off between safety and helpfulness is discussed in PrivacyLens (Sec 5.1) and 1-2-3 check. [1] introduces two metrics (i) leakage accuracy, (ii) benign succeeded, which jointly capture the trade-off (Sec 3.5 and Fig 4).
>
> Why evaluate helpfulness? As an example. John asks an LM agent to send a progress report to his manager. The LM agent may accidentally share information that John is “talking to a few companies about switching jobs” when assisting John in sending an email to John’s manager. Importantly, **the LM cannot simply refuse the request to avoid leakage**; it must still complete the task. A similar scenario in the congregant-clergy setting is illustrated in Figure 4 of our manuscript.
>
> [1] Patil, Vaidehi et.al. "The sum leaks more than its parts: Compositional privacy risks and mitigations in multi-agent collaboration." arXiv preprint arXiv:2509.14284.
>
>
> > W2. [Major] LC-ARS hyperparameter sensitivity unverified.
> >
> > Q1. Justification on the choice of α, β values
>
> Thank you for this insightful suggestion.
>
> To address this, we conducted additional ablations over a range of values, as shown in the table below. Specifically, we compare 6 different shaping component configurations of training on the base model Llama-3.1-8B-Instruct. The default choice in the paper is $\alpha = 0.5, \beta = 2.0$.
>
>
> | Method | Privacy Leak ↓ (avg) | Privacy Leak ↓ (leak@K) | Helpfulness ↑ (avg) | Helpfulness ↑ (bin) |
> | :--- | :---: | :---: | :---: | :---: |
> | Vanilla | 9.697 | 41.414 | 0.710 | 0.734 |
> | PPE | 9.495 | 39.394 | 0.690 | 0.729 |
> | AIFC | 21.111 | 54.545 | 0.834 | 0.878 |
> | Ours ($\alpha = 0.3, \beta = 2.0$) | 6.869 | 31.313 | 0.724 | 0.763 |
> | Ours ($\alpha = 0.5, \beta = 1.5$) | 6.364 | 32.323 | 0.713 | 0.751 |
> | Ours ($\alpha = 0.5, \beta = 2.0$) | 6.263 | 32.323 | 0.737 | 0.782 |
> | Ours ($\alpha = 1.0, \beta = 2.0$) | 8.586 | 35.354 | 0.734 | 0.767 |
> | Ours ($\alpha = 0.5, \beta = 3.0$) | 12.323 | 42.424 | 0.788 | 0.839 |
> | Ours ($\alpha = 2.0, \beta = 3.0$) | 9.495 | 37.374 | 0.766 | 0.811 |
>
> 1. Robustness across a reasonable range. Across different $\alpha$ and $\beta$, the results demonstrate robustness - all configurations except one strictly dominate Vanilla LM and PPE across both metrics. AIFC improves helpfulness only by substantially increasing leakage.
>
> 2. Effect of $\alpha$ (leakage shaping) and $\beta$ (helpfulness shaping).
>
> - Fixing $\beta=2.0$, increasing $\alpha$ (0.3→0.5→1.0) slightly increases leakage, confirming that $\alpha<1$ imposes a concave penalty that strongly discourages even small leakage.
> - Fixing $\alpha=0.5$, increasing $\beta$ (1.5→2.0→3.0) improves helpfulness, consistent with LC-ARS emphasizing utility with larger $\beta$.
>
> > Q2. Hyperparameter sensitivity
>
> $b_2$ serves as a control knob over the privacy-helpfulness tradeoff by determining the base reward for non-leaking behaviors. As shown in this table, increasing $b_2$ shifts the model from a helpfulness-oriented to a privacy-oriented regime.
>
> | Method | Privacy Leak ↓ (avg) | Privacy Leak ↓ (leak@K) | Helpfulness ↑ (avg) | Helpfulness ↑ (bin) |
> | :--- | :---: | :---: | :---: | :---: |
> | Ours ($b_2 = -0.5$) | 12.222 | 52.525 | 0.768 | 0.814 |
> | Ours ($b_2 = -0.2$) | 13.333 | 48.485 | 0.800 | 0.832 |
> | Ours ($b_2 = 0.0$)  | 12.121 | 47.475 | 0.822 | 0.869 |
> | Ours ($b_2 = 0.2$)  | 12.121 | 50.505 | 0.788 | 0.830 |
> | Ours ($b_2 = 0.3$)  | 9.596  | 44.444 | 0.775 | 0.821 |
> | Ours ($b_2 = 0.4$)  | 8.081  | 34.343 | 0.751 | 0.788 |
> | Ours ($b_2 = 0.45$) | 6.263  | 32.323 | 0.737 | 0.782 |
> | Ours ($b_2 = 0.5$)  | 4.545  | 21.212 | 0.638 | 0.660 |
>
> > W3. [Moderate] LLM-as-a-judge reliability warrants validation
> >
> > Q3. LLM judge consistency
>
> We conduct an additional validation using GPT-4o as judge, which leads to consistent conclusion:
>
> | Method | Privacy Leak ↓ (avg) | Privacy Leak ↓ (leak@K) | Helpfulness ↑ (avg) | Helpfulness ↑ (bin) |
> | :--- | :---: | :---: | :---: | :---: |
> | Vanilla | 13.535 | 49.495 | 0.613 | 0.622 |
> | PPE | 14.646 | 50.505 | 0.598 | 0.619 |
> | AIFC | 31.010 | 66.667 | 0.777 | 0.793 |
> | Ours ($b_2 = 0.0$) | 20.000 | 53.535 | 0.699 | 0.718 |
> | Ours ($b_2 = 0.3$) | 15.152 | 48.485 | 0.684 | 0.700 |
> | Ours ($b_2 = 0.45$) | 9.899 | 42.424 | 0.648 | 0.660 |
>
>
> > W4. [Minor]
>
> Thank you for pointing this out. We will explicitly acknowledge it in the discussion.
>
> > Q4. Reward propagation design
>
> Expected-value averaging is motivated by Monte Carlo estimation, where the mean of bounded rewards—by Hoeffding’s inequality—provides a high-probability, statistically consistent estimate of downstream utility.
>
> In contrast,
> minimum-value propagation introduces pessimistic bias;
> discounted propagation assumes additive stepwise rewards, which does not apply in our setting where rewards are only defined at the final node.
>
> We will clarify this design choice in the revision.

---

> > ### Author Rebuttal · Reviewer_nFzu · 2026-04-03
> >
> > Thanks for the rebuttal.

---

> > > ### Author Response · Authors · 2026-04-03
> > >
> > > We sincerely thank you for your time throughout the review process. We are glad that our rebuttal addressed your comments.

---

### Official Review · Reviewer_F2wq · 2026-03-12

**Soundness:** 3
**Presentation:** 3
**Significance:** 3
**Originality:** 4
**Overall Recommendation:** 5
**Confidence:** 4

**Summary:**

This paper explores how to enable large language model-based multi-agent systems to proactively protect contextual privacy. Existing methods typically rely on prompt engineering or external gatekeeper agents during the inference phase to filter sensitive information. However, such approaches are often fragile, hard to generalize, and can increase the attack surface by exposing intermediate reasoning processes.  To address these issues, the authors propose internalizing privacy preferences directly into the model's training process. They utilize a tree-structured multi-agent generation framework to collect preference data and employ reward backpropagation to assign training signals to each intermediate agent. Additionally, the authors design an asymmetric reward mechanism under the premise of zero privacy leakage to balance the trade-off between privacy protection and utility.  Experimental results demonstrate that this method effectively reduces privacy leakage rates across multiple models and benchmarks, with a maximum reduction of approximately 12%, while maintaining comparable task completion quality. Furthermore, the approach exhibits strong generalization capabilities across different agent topology structures.

**Compliance With Llm Reviewing Policy:**

Affirmed.

**Key Questions For Authors:**

1. The paper employs Mistral-7B-Instruct-v0.2 both as the evaluation backbone and as one of the models being assessed. Has the author considered utilizing an independent and more advanced evaluation model, such as GPT-4o, for cross-validation? Could self-assessment inherently lead to a systematic overestimation of the performance on the Mistral backbone?

2. LC-ARS establishes a rigid zero-privacy-leakage constraint as a prerequisite for utility rewards. In real-world scenarios, however, partial privacy leakage may be unavoidable (e.g., residual risks in obfuscated data). Could such a strict condition result in the model becoming overly conservative in complex scenarios, thereby causing unacceptable utility losses in practical applications?

3. The current framework assumes that sensitive information is pre-defined and independent. However, recent studies have highlighted that combining multiple seemingly innocuous pieces of information can lead to emergent privacy breaches. Is PrivAct equipped to handle such combinatorial privacy risks, or does it still rely on manually curated sets of sensitive information?

**Limitations:**

yes

**Strengths And Weaknesses:**

**Strengths：**
- By internalizing privacy preferences into model weights, this approach avoids the attack surface expansion caused by lengthy prompts during inference, such as CoT reasoning trace leakage, providing a more fundamental solution compared to external interventions.

- In the privacy-utility trade-off space, PrivAct establishes a superior Pareto frontier across all tested backbone models, rather than simply trading utility for privacy.

- Zero-shot transfer results on ConfAIde demonstrate that privacy capabilities trained on action tasks can transfer to detection tasks, indicating that the model has acquired a deeper understanding of privacy rather than task-specific patterns.

- The LC-ARS mechanism, through asymmetric exponentiation and conditional design, effectively addresses "shortcut" issues in multi-objective optimization.

- The approach maintains performance advantages across six different multi-agent topologies without requiring retraining for specific structures, showcasing strong engineering practicality.

**Weaknesses：**
- The PrivacyLens test set contains only 99 samples, raising concerns about statistical significance and its ability to adequately capture the diversity and long-tail distribution of real-world privacy scenarios.

- Both privacy leakage and utility are evaluated using Mistral-7B as the LLM-as-a-judge, while Mistral-7B itself is one of the backbone models being evaluated, introducing a risk of self-assessment bias.

- The improvement on Llama-1B is minimal (leakage rate reduced from 12.02% to just 10.51%), indicating that the method may rely on model capacity and its effectiveness diminishes in resource-constrained settings.

---

> ### Author Rebuttal · Authors · 2026-03-30
>
> > W1. The PrivacyLens test set contains only 99 samples...
>
> We thank the reviewer for raising this concern.
> While the PrivacyLens, after splitting into train/test, contains 99 test samples, it is curated to maximize diversity. PrivacyLens employs a multi-sourced generation pipeline that results in domain, role, and interaction diversity.
>
> To further address concerns about scale and generalization, we complement our evaluation with another 2 independent benchmarks.
> - ConfAIde, including Tier-3 (**270** scenarios) and Tier-4 (**20** scenarios), with results reported in Sec. 4.3 of our paper.
> - A compositional privacy leakage dataset [1], which consists of **119** scenarios. The results are provided under our response to Q3.
>
> > W2. using Mistral-7B as the LLM-as-a-judge...
> >
> > Q1. The paper employs Mistral-7B-Instruct-v0.2 ... Has the author considered utilizing an independent and more advanced evaluation model, such as GPT-4o, for cross-validation? ...
>
> Thank you for this important point. We conduct an additional validation using GPT-4o as judge on the mistral backbone:
>
> | Method | Privacy Leak ↓ (avg) | Privacy Leak ↓ (leak@K) | Helpfulness ↑ (avg) | Helpfulness ↑ (bin) |
> | :--- | :---: | :---: | :---: | :---: |
> | Vanilla | 28.788 | 65.657 | 0.708 | 0.724 |
> | PPE | 28.283 | 62.626 | 0.688 | 0.703 |
> | AIFC | 37.273 | 75.758 | 0.780 | 0.787 |
> | Ours | 24.646 | 60.606 | 0.695 | 0.705 |
>
> Gpt-4o's judge results produces a consistent conclusion: our method achieves the lowest privacy leakage while maintaining competitive helpfulness.
>
> > W3. The improvement on Llama-1B is minimal (leakage rate reduced from 12.02% to just 10.51%) ...
>
> We thank the reviewer for this insightful observation.
> The absolute leakage rate reduction on Llama-1B (from 12.02% to 10.51%) appears modest. However, this comparison should be considered jointly with the helpfulness improvement, which increases 2.7% (0.368 to 0.395). Under a fixed-helpfulness setting, the achievable leakage reduction would be larger.
> We also acknowledge that the model capacity plays a role: larger models exhibit stronger improvements, likely due to their greater representational capacity to balance complex privacy and helpfulness trade-offs. We will include additional discussion to clarify this point.
>
> > Q2. LC-ARS establishes a rigid zero-privacy-leakage constraint as a prerequisite for utility rewards. ... Could such a strict condition result in the model becoming overly conservative in complex scenarios, thereby causing unacceptable utility losses in practical applications?
>
> Thank you for this thoughtful question.
> This LC-ARS design establishes this constraint to discourage the shortcut of trading small privacy violations for large utility gains, thereby encouraging the model to explore strategies that achieve high utility without leaking sensitive information.
>
> At the same time, LC-ARS provides flexibility in tuning the privacy-helpfulness balance. As shown in Fig 2 of our paper, varying $b_2$ produces a spectrum of operating points along the privacy-helpfulness frontier. This allows adaptation to different application requirements, depending on how costly privacy leakage is in the specific scenario.
>
> > Q3. ... recent studies have highlighted that combining multiple seemingly innocuous pieces of information can lead to emergent privacy breaches. Is PrivAct equipped to handle such combinatorial privacy risks? ...
>
> A recent work [1] studies compositional privacy leakage where adversaries can recover sensitive information by composing innocuous responses, even when each response is benign in isolation. For example, an attacker may separately obtain customer ID–name mappings, product purchase logs, and insurance claim information from different agents. While no single response is harmful in isolation, their combination reveals that “John, with no diagnosed heart condition, is self-monitoring for potential undiagnosed heart issues.”
>
> Their dataset consists of 119 scenarios, each evaluated under both adversarial and benign inferences. We compare the baseline Chain-of-Thought (CoT) defenses with variants where the defer agents are replaced by models trained with PrivAct. The defender's base model is Llama-3.1-8B-Instruct and the attacker is Qwen3-32B.
>
> Evaluation is conducted using the Overall Success metric defined in [1], which measures the percentage of scenario pairs where the benign query succeeds and the corresponding sensitive query is fully blocked. Across different parameters settings of $b_2$, and for both verifier and refiner models, our method consistently outperforms the baseline.
>
> | Method | Overall Succ. |
> | :--- | :--- |
> | CoT | 42.39 |
> | CoT+Ours (verifier, $b_2=0.3$) | 44.3 |
> | CoT+Ours (refiner, $b_2=0.3$) | 46.34 |
> | CoT+Ours (refiner, $b_2=0.45$) | 45.35 |
>
>
> [1] Patil, Vaidehi et.al. "The sum leaks more than its parts: Compositional privacy risks and mitigations in multi-agent collaboration." arXiv preprint arXiv:2509.14284.

---

> > ### Author Rebuttal · Reviewer_F2wq · 2026-04-02
> >
> > The authors have provided detailed responses to all my questions and supplemented their answers with comprehensive experimental results.

---

> > > ### Author Response · Authors · 2026-04-03
> > >
> > > We sincerely thank the reviewer for recognizing the effort put into additional experiments. We are glad that our responses and new results addressed your questions.

---

### Official Review · Reviewer_9xgn · 2026-03-12

**Soundness:** 3
**Presentation:** 2
**Significance:** 3
**Originality:** 3
**Overall Recommendation:** 5
**Confidence:** 3

**Summary:**

This paper proposes PrivAct, a multi-agent preference learning framework that internalizes contextual privacy preservation into the generation behavior of language model agents. The manuscript’s key aspect is shifting from inference-time privacy enforcement (e.g., prompt engineering or external gatekeepers) to training-time internalization via reward propagation and leakage-conditioned asymmetric reward shaping (LC-ARS). The authors aim to explore an important domain: ensuring contextual integrity in agentic LLM systems without expanding the attack surface through explicit reasoning traces. Empirical results across multiple backbones and benchmarks show improved privacy–helpfulness tradeoffs and robustness across agent topologies.

**Compliance With Llm Reviewing Policy:**

Affirmed.

**Final Justification:**

The authors’ rebuttal has addressed my concerns, and I am raising my score from 4 to 5. I hope the authors will incorporate the content of the rebuttal into the paper.

**Key Questions For Authors:**

As shown in the weaknesses.

**Limitations:**

yes

**Strengths And Weaknesses:**

**Strengths**
* S1. The paper makes a meaningful conceptual shift by embedding contextual privacy constraints directly into generation policies rather than enforcing them externally. This internalization perspective is novel within multi-agent LLM systems and well-motivated.
* S2. The proposed LC-ARS mechanism is technically elegant and addresses the structural privacy–utility tradeoff. By conditioning helpfulness optimization on zero leakage, the framework treats privacy as a prerequisite instead of a competing objective. This design is both principled and empirically validated.
* S3. The reward propagation strategy enables per-agent preference construction in a structured manner. This is a practical and generalizable contribution to multi-agent fine-tuning, ensuring alignment across intermediate agents rather than solely optimizing final outputs.

**Weaknesses**
* W1. The framework depends on an LLM-based judge to compute leakage signals used in reward shaping. The robustness, calibration, and potential bias of this judge are not thoroughly analyzed. Given that reward quality directly impacts preference learning, further validation of the judging mechanism would strengthen the work.
* W2. The LC-ARS formulation introduces multiple shaping parameters. While some configurations are explored to trace privacy–helpfulness frontiers, the paper lacks a systematic sensitivity analysis. It is unclear how stable performance is across different parameter choices or whether tuning is required per backbone.
* W3. Experiments are conducted primarily on structured benchmarks PrivacyLens and ConfAIde. It remains unclear how well the approach generalizes to more open-ended or production-level agent systems with less controlled interaction patterns.

---

> ### Author Rebuttal · Authors · 2026-03-30
>
> > W1. The robustness, calibration, and potential bias of this judge are not thoroughly analyzed
>
> Thank you for this important point. To assess the validity and robustness of the LLM-judge, we conducted additional analyses:
>
> 1. Agreement with human annotation
>
> We adopt the same LM judge (Mistral-7B-Instruct-v0.2) as used in PrivacyLens, where its reliability has been validated against human annotators. Specifically, PrivacyLens reports results (in Sec 5.1) from 4 human annotators on 153 (action, sensitive information) pairs for leakage detection and 50 actions for helpfulness evaluation. For privacy leakage, the agreement between the LLM judge and human annotators achieves a Fleiss’ Kappa of 0.82, with an accuracy of 0.92 when treating human labels as ground truth. For helpfulness evaluation, the agreement is 0.56 with Fleiss’ Kappa.
>
> 2. Cross-model consistency
>
> We evaluate the same outputs using GPT-4o as the judge and find that the results are consistent with results using Mistral-7B-Instruct-v0.2 as the judge:
>
> | Method | Privacy Leak ↓ (avg) | Privacy Leak ↓ (leak@K) | Helpfulness ↑ (avg) | Helpfulness ↑ (bin) |
> | :--- | :---: | :---: | :---: | :---: |
> | Vanilla | 13.535 | 49.495 | 0.613 | 0.622 |
> | PPE | 14.646 | 50.505 | 0.598 | 0.619 |
> | AIFC | 31.010 | 66.667 | 0.777 | 0.793 |
> | Ours ($b_2 = 0.0$) | 20.000 | 53.535 | 0.699 | 0.718 |
> | Ours ($b_2 = 0.3$) | 15.152 | 48.485 | 0.684 | 0.700 |
> | Ours ($b_2 = 0.45$) | 9.899 | 42.424 | 0.648 | 0.660 |
>
> Ours achieves the lowest privacy leakage rate while maintaining a competitive level of helpfulness. In contrast, prompt-based privacy enhancement (PPE) and agent-based information flow control (AIFC) exhibit a worse tradeoff.
>
> We will include these additional validations and a more detailed discussion of the LLM judge's validity and robustness in the revised version to further strengthen the reliability of our evaluation framework.
>
>
> > W2. The LC-ARS formulation introduces multiple shaping parameters. While some configurations are explored to trace privacy–helpfulness frontiers, the paper lacks a systematic sensitivity analysis...
>
> Thank you for this insightful suggestion.
> We include a sensitivity analysis:
>
> | Method | Privacy Leak ↓ (avg) | Privacy Leak ↓ (leak@K) | Helpfulness ↑ (avg) | Helpfulness ↑ (bin) |
> | :--- | :---: | :---: | :---: | :---: |
> | Vanilla | 9.697 | 41.414 | 0.710 | 0.734 |
> | PPE | 9.495 | 39.394 | 0.690 | 0.729 |
> | AIFC | 21.111 | 54.545 | 0.834 | 0.878 |
> | Ours ($\alpha = 0.3, \beta = 2.0$) | 6.869 | 31.313 | 0.724 | 0.763 |
> | Ours ($\alpha = 0.5, \beta = 1.5$) | 6.364 | 32.323 | 0.713 | 0.751 |
> | Ours ($\alpha = 0.5, \beta = 2.0$) | 6.263 | 32.323 | 0.737 | 0.782 |
> | Ours ($\alpha = 1.0, \beta = 2.0$) | 8.586 | 35.354 | 0.734 | 0.767 |
> | Ours ($\alpha = 0.5, \beta = 3.0$) | 12.323 | 42.424 | 0.788 | 0.839 |
> | Ours ($\alpha = 2.0, \beta = 3.0$) | 9.495 | 37.374 | 0.766 | 0.811 |
>
> 1. Robustness across a reasonable range. Across different $\alpha$ and $\beta$, the results demonstrate robustness - all configurations except one strictly dominate Vanilla LM and PPE across both metrics. AIFC improves helpfulness only by substantially increasing leakage.
>
> 2. Effect of $\alpha$ (leakage shaping) and $\beta$ (helpfulness shaping).
>
> - Fixing $\beta=2.0$, increasing $\alpha$ (0.3→0.5→1.0) slightly increases leakage, confirming that $\alpha<1$ imposes a concave penalty that strongly discourages even small leakage.
> - Fixing $\alpha=0.5$, increasing $\beta$ (1.5→2.0→3.0) improves helpfulness, consistent with LC-ARS emphasizing utility with larger $\beta$.
>
> > W3. Experiments are conducted primarily on structured benchmarks PrivacyLens and ConfAIde. It remains unclear how well the approach generalizes to more open-ended or production-level agent systems with less controlled interaction patterns.
>
> Thank you for raising this concern.
> To address this, we included additional evaluation on a recent benchmark studying compositional privacy risks in multi-agent systems [1]. It features less controlled interaction patterns, where privacy risks emerge from multi-step, multi-agent interactions and information composition rather than single-turn decisions.
>
> Evaluation is conducted using the Overall Success metric defined in [1], which measures the percentage of scenario pairs where the benign query succeeds and the corresponding sensitive query is fully blocked. Across different parameters settings of $b_2$, and for both verifier and refiner models, our method consistently outperforms the baseline.
>
> | Method | Overall Succ. |
> | :--- | :--- |
> | CoT | 42.39 |
> | CoT+Ours (verifier, $b_2=0.3$) | 44.3 |
> | CoT+Ours (refiner, $b_2=0.3$) | 46.34 |
> | CoT+Ours (refiner, $b_2=0.45$) | 45.35 |
>
> We will include these additional results and discussions in the revision.
>
> [1] Patil, Vaidehi, Elias Stengel-Eskin, and Mohit Bansal. "The sum leaks more than its parts: Compositional privacy risks and mitigations in multi-agent collaboration." arXiv preprint arXiv:2509.14284.

---

> > ### Author Rebuttal · Reviewer_9xgn · 2026-04-02
> >
> > Thank you for your valuable comment. My question has now been resolved.

---

> > > ### Author Response · Authors · 2026-04-03
> > >
> > > We sincerely thank the reviewer for the feedback. We are pleased to hear that our responses addressed your questions.

---

### Decision · Program_Chairs · 2026-04-30

**Decision:**

Accept (regular)

**Comment:**

This paper studies contextual privacy preservation in multi-agent LLM systems and proposes PrivAct, a training-based framework that internalizes privacy-aware behavior through multi-agent preference learning and leakage-conditioned asymmetric reward shaping. Reviewers agreed that the paper tackles an important problem and presents a technically meaningful alternative to inference-time privacy controls.

After considering the reviews, rebuttal, and discussion, I find the paper strong enough for acceptance. Reviewers found the core idea novel and practically relevant, and the experiments show consistent gains in privacy-helpfulness tradeoffs across multiple backbones, transfer settings, and agent topologies. The rebuttal further strengthened the paper by adding judge-validation experiments, sensitivity analyses, and additional evaluation on compositional privacy settings.

The paper is not without limitations. Reviewers noted the dependence on LLM-as-a-judge, some deployment constraints, and sensitivity to hyperparameter choices. However, these concerns were addressed reasonably in rebuttal and do not outweigh the overall contribution.

Overall, this is a solid and timely contribution to privacy-aware agentic systems.